# p75NTR prevents the onset of cerebellar granule cell migration via RhoA activation

**Juan P Zanin\*, Wilma J Friedman**

Department of Biological Sciences, Rutgers University, Newark, United States

**Abstract** Neuronal migration is one of the fundamental processes during brain development. Several neurodevelopmental disorders can be traced back to dysregulated migration. Although substantial efforts have been placed in identifying molecular signals that stimulate migration, little is known about potential mechanisms that restrict migration. These restrictive mechanisms are essential for proper development since it helps coordinate the timing for each neuronal population to arrive and establish proper connections. Moreover, preventing migration away from a proliferative niche is necessary in maintaining a pool of proliferating cells until the proper number of neuronal progenitors is attained. Here, using mice and rats, we identify an anti-migratory role for the p75 neurotrophin receptor (p75NTR) in cerebellar development. Our results show that granule cell precursors (GCPs) robustly express p75NTR in the external granule layer (EGL) when they are proliferating during postnatal development, however, they do not express p75NTR when they migrate either from the rhombic lip during embryonic development or from the EGL during postnatal development. We show that p75NTR prevented GCP migration by maintaining elevated levels of active RhoA. The expression of p75NTR was sufficient to prevent the migration of the granule cells even in the presence of BDNF (brain-derived neurotrophic factor), a well-established chemotactic signal for this cell population. Our findings suggest that the expression of p75NTR might be a critical signal that stops and maintains the GCPs in the proliferative niche of the EGL, by promoting the clonal expansion of cerebellar granule neurons.

**\*For correspondence:**
jz332@newark.rutgers.edu

**Competing interest:** The authors declare that no competing interests exist.

## Editor's evaluation

This study investigates a key molecular mechanism that drives neuronal migration. The results indicate that the p75 neurotrophin receptor provides an anti-migratory cue for granule cell precursors in developing cerebellum. This topic has been a topic of wide developmental interest that integrates previous and recent findings.

## Introduction

Neuronal migration is a fundamental process in nervous system development. Several neurological disorders can be traced back to defects in migration during development (*Buchsbaum and Cappello, 2019*). In mammals, a highly coordinated series of neuronal migration events are essential for the establishment of the layered structure of the brain, while bringing neurons into the correct location to assemble the proper neuronal connections.

The cerebellum is well known for its role in motor control, yet it is also involved in non-motor functions such as decision making (*Deverett et al., 2018*), reward anticipation (*Wagner et al., 2017*), and social interaction (*Carta et al., 2019*). Cerebellar granule cells are a unique neuronal population since they undergo two rounds of proliferation/migration cycle during development. In the first round, starting at E10.5 in rats, cells proliferate in the rhombic lip, and around E12.5 the granule cell precursors (GCPs) begin to migrate tangentially on the surface of the future cerebellum, where they will stop migrating and establish a transient proliferative zone, the external granule layer (EGL) (*Altman*

**eLife digest** The human brain contains billions of neurons that form vast networks to relay information around the brain and to the rest of the body. The numbers and locations of neurons, and the connections between them, affect how the brain works, so the body carefully controls how, where and when neurons form.

Most of the neurons in the brain arise before we are born from groups of supporting cells known as neuronal precursors. Often, these cells must migrate from one place to another to make neurons in the correction location. For example, neuronal precursors in an area of the embryo brain, called the rhombic lip, produce granule cells – a type of neuron found in the cerebellum, a region of the adult brain that controls our ability to move around. Before making the neurons, the precursor cells first have to migrate out of the rhombic lip into a neighboring area.

Previous studies indicate that a protein known as p75NTR may help to control the ability of brain cells to migrate, but its precise role remained unclear. To address this question, Zanin and Friedman investigated the role of p75NTR in the migration of granule cell precursors in mice and rats.

The experiments found that in animals lacking this protein, the granule cell precursors began to migrate out of the rhombic lip earlier than in normal animals, resulting in excessive numbers of granule cells in the adult cerebellum, which can affect the normal development of an animal. The p75NTR protein appeared to prevent the cells from migrating by activating another protein called RhoA.

Understanding how the body controls when neuronal precursors and other brain cells migrate helps us to understand how the brain develops in healthy individuals and certain neurological disorders, including autism. The next step is to find out whether p75NTR also plays a similar role in the human brain.

*and Bayer, 1996*). In the EGL, the GCPs start the second round of proliferation, undergoing clonal expansion during the first 2–3 postnatal weeks. During this entire period, waves of GCPs exit the cell cycle and start the second migratory stage. Using the Bergman glia as a track together with molecular cues, the GCPs pass through the Purkinje cell layer and colonize the internal granule layer (IGL), their final destination. Although several studies have identified signals that stimulate and guide granule cell migration, little is known about potential mechanisms that prevent the onset of migration. These restrictive mechanisms are critical for proper development since they help to coordinate the proper timing of cell migration away from a proliferative niche such as the EGL (*Altman and Bayer, 1996*).

The p75 neurotrophin receptor (p75NTR) can mediate a broad array of functions depending on the cellular context, including stimulation or inhibition of cell migration. For instance, p75NTR can stimulate the chemotactic migration of neural crest cells (*Zanin et al., 2013*). P75NTR stimulates the migration of neuroblasts from the subventricular zone into the olfactory bulb (*Snapyan et al., 2009*). This receptor also promotes cell migration in certain types of cancers, such as glioblastoma (*Berghoff et al., 2015*; *Wang et al., 2015*) and melanoma (*Shonukan et al., 2003*; *Truzzi et al., 2008*). In contrast, the receptor has also been implicated in the inhibition of Schwann cell migration in a mechanism dependent on RhoA activation (*Yamauchi et al., 2004*). Recently, p75NTR has been implicated in preventing the aberrant migration of granule cells into the white matter of the IGL in the cerebellum by triggering apoptosis, in a mechanism involving myelin-associated glycoprotein (*Fernández-Suárez et al., 2019*). These contradictory results emphasize the complexity of p75NTR signaling and functions, re-enforcing the necessity of studying the role of this receptor in specific cell contexts to fully understand its biological functions. During cerebellar development, p75NTR is highly expressed in the proliferating GCPs located in the outer EGL and is downregulated before the GCPs start migrating (*Carter et al., 2003*; *Zanin et al., 2016*). Among the multiple signaling proteins that are regulated by p75NTR, RhoA regulation is particularly interesting, since p75NTR can maintain elevated levels of active RhoA in a ligand-independent manner (*Yamashita et al., 1999*; *Yamashita and Tohyama, 2003*). RhoA is a member of the small sub-family of Rho-GTPases that cycle between a GTP-bound (active) and a GDP-bound (inactive) form. Through this mechanism, these molecules can coordinate multiple aspects of cellular responses. In particular, due to their regulation of cytoskeletal dynamics, these GTPases are involved in different aspects of neuronal migration (*Govek et al., 2011*

and *Govek et al., 2018*; *Xu et al., 2019*). Therefore, disruption of Rho-GTPase activity has been associated with migration defects leading to several behavioral and developmental disorders (*Boettner and Van Aelst, 2002*; *Govek et al., 2005* and *Govek et al., 2018*). Regulation of this signaling pathway suggests the possibility that p75NTR can influence neuronal migration via the regulation of RhoA activity.

In the present study, we aimed to investigate the role of p75NTR in regulating cerebellar granule neuron (CGN) migration. We found that this receptor exerts an anti-migratory effect on the granule cells, both during embryonic and postnatal development, and must be downregulated to allow the CGNs to migrate away from the EGL. Ex vivo analysis showed that overexpression of either p75NTR or activated RhoA blocks the migration of granule cells. Our findings reveal a novel role for p75NTR in preventing neuronal migration by maintaining RhoA activation in the developing cerebellum.

## Results

### P75NTR is expressed in mitotic, non-migrating GCPs

During early embryonic development (E13.5) the precursors of the granule cells, indicated by the expression of PAX6 (*Figure 1A-B* and *Figure 1—figure supplement 1*, magenta), begin to disperse rostrally from the rhombic lip. Interestingly, these precursor cells do not express the neuronal migratory marker DCX (*Figure 1A–B* and *Figure 1—figure supplement 1*, white), suggesting these cells did not yet acquire a neuronal fate. During this period, no expression of p75NTR was observed in the migrating cells (*Figure 1A–B* and *Figure 1—figure supplement 1*, green). At later stages (E15.5) the majority of the precursor cells still migrate tangentially to complete the formation of the EGL. However, a subpopulation of cells begins to emerge, located in the internal part of the migratory stream, these cells express PAX6 (*Figure 1A–B* and *Figure 1—figure supplement 1*, magenta) and the neurotrophin receptor p75NTR (*Figure 1A–B* and *Figure 1—figure supplement 1*, green). At the final stages of embryonic development (E17.5), there is an enlargement of the EGL, which is accompanied by an increase in the number of p75NTR expressing cells (*Figure 1A–B* and *Figure 1—figure supplement 1*, green), which continues throughout postnatal development until the EGL disappears between 2 and 3 weeks after birth. During postnatal development, p75NTR is expressed in the GCPs throughout the entire EGL (*Figure 1C–F*, green). Moreover, the expression of p75NTR closely correlates with the proliferation levels in the EGL, with a peak of p75NTR expression during the maximal proliferation stage of the GCPs, which in mice occurs around P5–7 (*Figure 1C*). Postmitotic GCPs, located mostly in the internal part of the EGL, begin to express neuronal differentiation and migratory markers such as TAG1 (*Figure 1D*, magenta) or DCX (*Figure 1E*, magenta) in preparation for the radial migratory phase toward the IGL. In contrast to the expression of p75NTR in proliferating cells, postmitotic cells downregulate the expression of p75NTR, and no overlap between p75NTR and the migratory markers was observed (*Figure 1D–F*). This well-defined boundary between the proliferating cells expressing p75NTR and the migrating TAG1 or DCX positive cells indicates that exiting the cell cycle is accompanied by a reduction in p75NTR expression in GCPs as they leave the cell cycle. The negative correlation between the cells that express p75NTR and the migratory cells observed at embryonic as well as postnatal stages suggests that the expression of the receptor could be a signal that stops the migration of cells coming from the rhombic lip, and retains the cells in the EGL until they are ready to migrate radially toward the IGL.

### Shh induces GCP proliferation and maintains a high expression of p75NTR

Although there is a clear transition between the proliferating cells expressing p75NTR and the differentiating cells that downregulate the expression of the receptor, the specific function of the receptor in this transition is not clear. For instance, it is unclear whether p75NTR is required to keep the GCPs in the cell cycle, or whether the cells maintain the expression of p75NTR because they are in the cell cycle.

To address this question, we took advantage of the spontaneous differentiation that the GCPs undergo in the absence of a mitogen in culture. In the presence of Shh, a well-known mitogen for GCPs (*Wallace, 1999*; *Kenney and Rowitch, 2000*), these cells are maintained in a proliferative and undifferentiated state with high expression levels of the proliferation marker PCNA (*Figure 2A, B*). However,

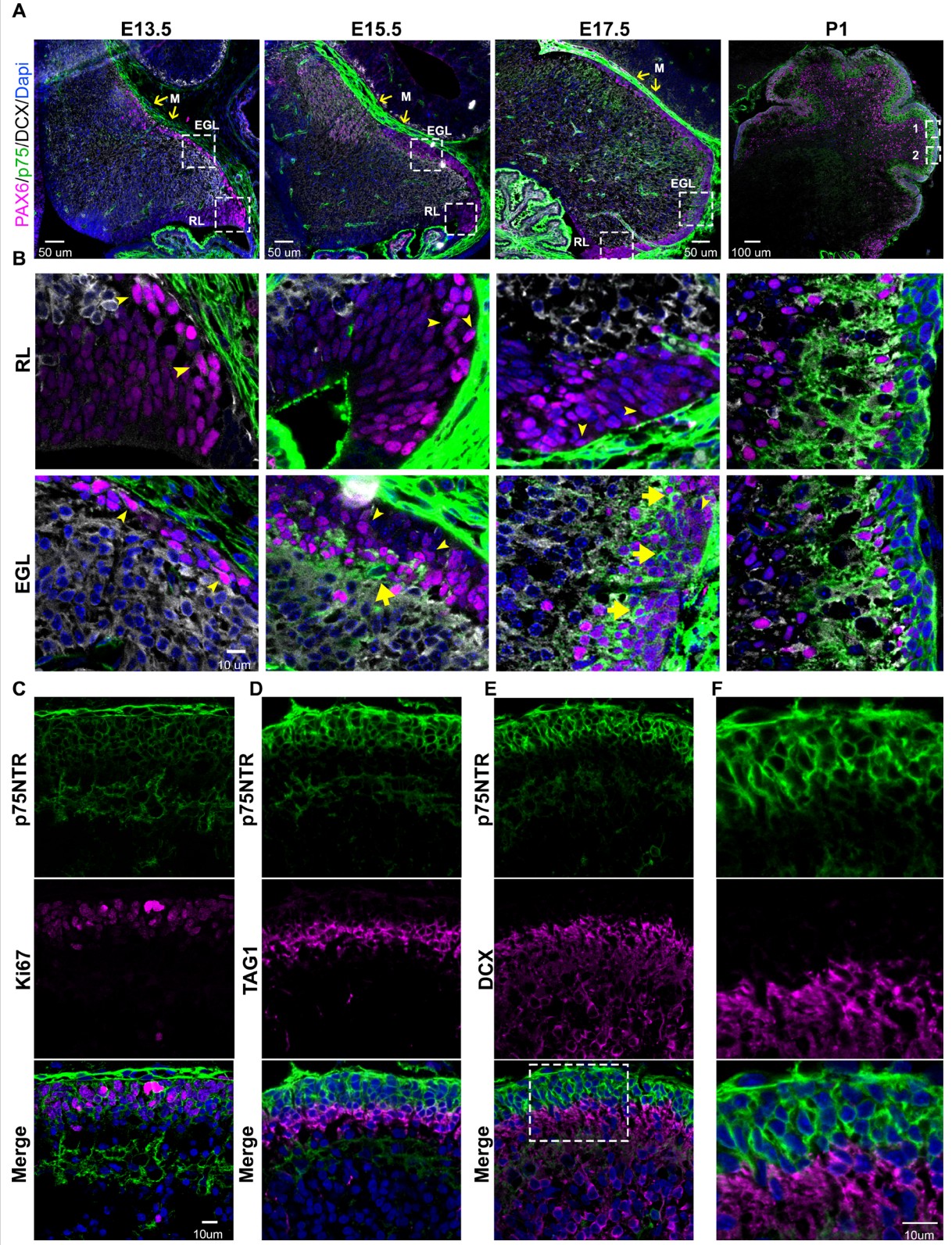

**Figure 1.** Developmental expression of p75 neurotrophin receptor (p75NTR) in mouse cerebellum. (**A**) Mouse developmental expression of p75NTR (green), Pax6 (magenta), DCX (white), and Dapi (blue) at the indicated ages. Note the high level of p75NTR in the meninges as well as the developing granule cell progenitors. Yellow arrows indicate the M – meninges, RL – rhombic lip, EGL – external granule layer. (**B**) High magnification of the insets showed in A. Cells expressing p75NTR (arrows), migrating cells negative for p75NTR (arrowheads). (**C**) Expression of p75NTR (green), Ki67 (magenta),

*Figure 1 continued on next page*

*Figure 1 continued*

and Dapi (blue) in the cerebellum of P7 mouse pups. (**D**) Immunohistochemistry of the expression of p75NTR (green), TAG1 (magenta), and Dapi (blue) in the cerebellum of P7 mouse pups. (**E**) Immunohistochemistry of the expression of p75NTR (green), DCX (magenta), and Dapi (blue) in the cerebellum of P7 mouse pups. (**F**) High magnification of the inset showed in E. The tissue shown in all the figures were obtained from mice.

The online version of this article includes the following figure supplement(s) for figure 1:

**Figure supplement 1.** Developmental expression of p75 neurotrophin receptor (p75NTR) in mouse cerebellum.

as early as 24 hr in culture without Shh, there is a significant reduction in the level of PCNA. We also observed that p75NTR expression significantly decreased in conditions without Shh, but remained elevated in the proliferative environment induced by Shh (***Figure 2A and C***). To further explore this positive correlation between proliferation levels and p75NTR expression, we exposed GCP cultures to different concentrations of Shh for 48 hr (***Figure 2D–K***). Shh induced a dose-dependent increase in proliferation, estimated by the number of Ki67+ cells (***Figure 2G*** top panels, 2H) and the relative increase in PCNA expression (***Figure 2D and E***). At the same time, Shh maintained high expression levels of p75NTR in a dose-dependent manner (***Figure 2G*** middle panels, 2D, F, J). Consistent with the observations in the EGL during cerebellar development (***Figure 1***), the number of double-labeled cells expressing p75 and Ki67 also increased in response to Shh in a dose-dependent manner (***Figure 2I***). Furthermore, all the proliferating cells also expressed p75NTR, even in the control conditions where the few remaining proliferating cells also expressed p75NTR (***Figure 2K***).

## Cell cycle exit correlates with a reduction in p75NTR levels

Previously, we have shown that proNT-3 can induce cell cycle exit of GCPs via p75NTR, even in the presence of Shh (***Zanin et al., 2016***). To determine whether inducing cell cycle exit of GCPs also elicits downregulation of p75NTR, we treated the cells with proNT-3 in the presence of the Shh analog SAG. Confirming the cell cycle withdrawal induced by proNT-3, we observed a reduction in the Ki67+ cells, which was accompanied by a significant decrease in p75NTR expression (***Figure 3A–C***, ***Figure 3— figure supplement 1***). Consistent with increased differentiation, proNT-3 induced an increase in the levels of βIII-tubulin even in the presence of SAG (***Figure 3A and D***, ***Figure 3—figure supplement 1***). To determine whether other signals that promote cell cycle withdrawal of GCPs also trigger a decrease in p75NTR expression, we treated granule cells with PACAP, a known anti-mitogenic ligand for GCPs (***Nicot et al., 2002***). We cultured GCPs for 48 hr in the presence of SAG with and without PACAP and quantified their proliferation either by Ki67 staining or PCNA levels using Western blots (***Figure 3E–K***, ***Figure 3—figure supplement 1***). PACAP induced a dose-dependent reduction in Ki67 (***Figure 3E and F***, ***Figure 3—figure supplement 1***) and PCNA levels (***Figure 3I–J***) and a concomitant reduction in p75NTR expression in a dose-dependent manner, even in the presence of SAG (***Figure 3E, G, I, K***). The reduced proliferation induced by PACAP was accompanied by increased neuronal differentiation, indicated by an increase in the levels of βIII-tubulin (***Figures 3E, H and 2***).

## P75NTR overexpression does not promote cell cycle re-entry

Our data have demonstrated a positive correlation between the expression of p75NTR and the proliferating GCPs, and a complete absence of the receptor upon cell cycle exit. To directly test whether the expression of p75NTR is sufficient to maintain the GCPs in a state permissive for proliferation even after withdrawal of Shh, which would allow the cells to re-enter the cell cycle upon re-addition of Shh, we transfected GCPs with a p75-GFP construct to maintain expression of the receptor even after Shh was withdrawn (***Figure 4***, ***Figure 4—figure supplement 1***). GCPs were then cultured in the absence of Shh for 24 hr, which is sufficient to reduce the levels of proliferation and the endogenous expression of p75NTR (see ***Figure 2***). The Shh analog SAG was then added for 24 hr and the proliferation levels were evaluated after the total 48 hr in culture. If p75NTR expression was sufficient to maintain the cells in a permissive proliferative state, we would expect to see an increase in proliferation with the re-exposure to Shh. As expected, cells that were maintained continuously in the presence of SAG for 48 hr continued to proliferate, regardless of the transfected construct (i.e. Ctrl-GFP or p75-GFP; ***Figure 4A*** right panels, 4B-F, ***Figure 4—figure supplement 1***) similar to the response observed in non-transfected cells (***Figures 2 and 4D–F***). Similarly, both transfected and non-transfected cells cultured in the absence of the mitogen for 48 hr did not proliferate (***Figure 4A*** left panels, 4B–F,

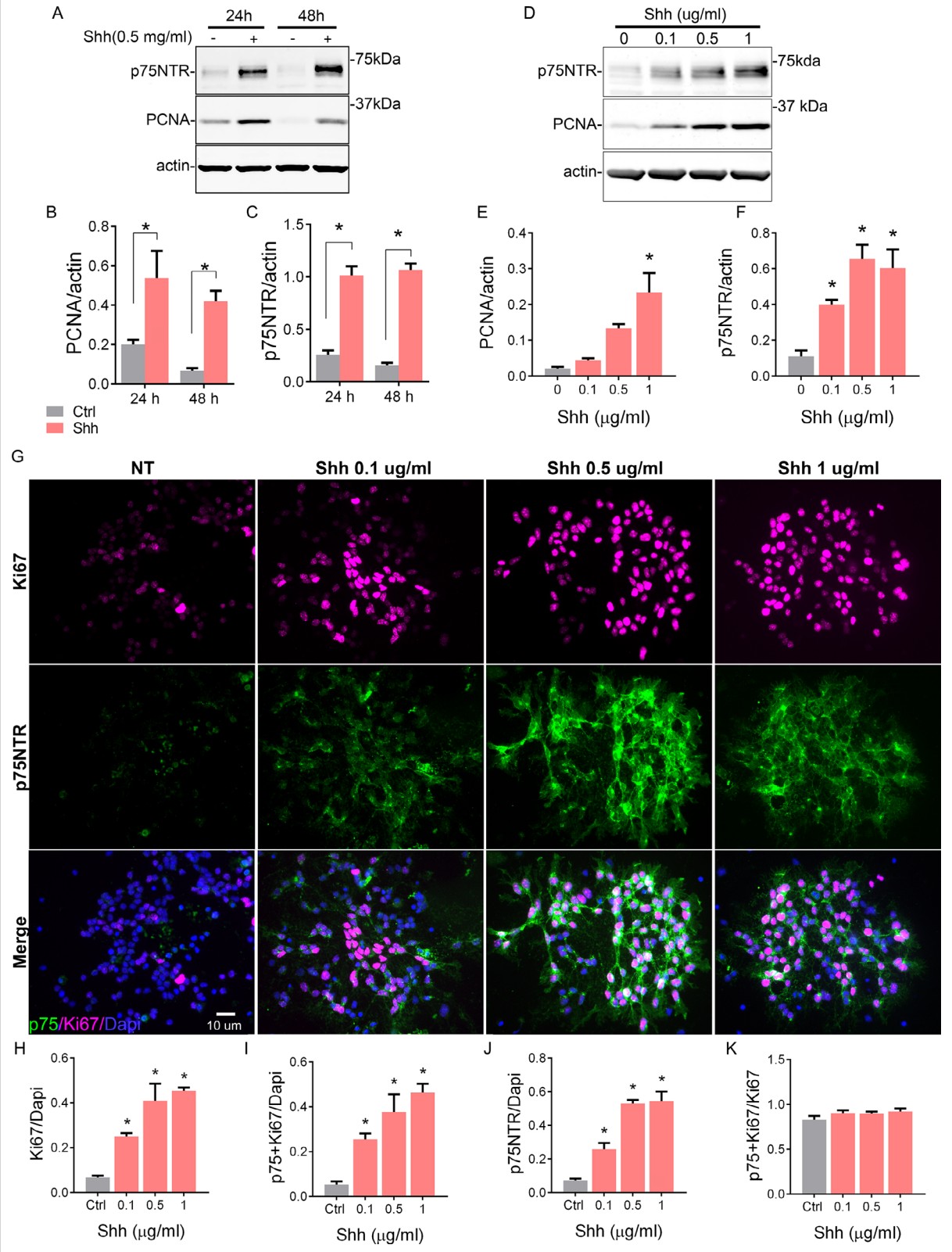

**Figure 2.** Shh regulates cerebellar granule neuron (CGN) proliferation and the expression of p75 neurotrophin receptor (p75NTR). (**A**) Western blot analysis of the temporal and dose-dependent expression of p75NTR and PCNA (proliferation marker) in granule cell cultures exposed to Shh. (**B and C**) Quantification of PCNA and p75NTR expression relative to actin. Two-way ANOVA, N=4, PCNA *p=0.0006, p75NTR *p=0.0001, error bars indicate SEM. (**D**) Western blot analysis of the expression of p75NTR and PCNA in granule cell cultures in response to increasing concentrations of Shh for

*Figure 2 continued on next page*

Figure 2 continued

48 hr. (**E** and **F**) Quantification of PCNA and p75NTR expression relative to actin. One-way ANOVA, N=3, PCNA *p=0.0026, p75NTR *p=0.0019, error bars indicate SEM. (**G**) Immunocytochemistry analysis of the expression of p75NTR (green), Ki67 (magenta), and Dapi (blue) in granule cell cultures in response to increasing concentrations of Shh for 48 hr. (**H**) Quantification of the total number of proliferating cells expressed as the percentage of cells expressing Ki67 over the total number of cells ANOVA, *p=0.0005, N=3, error bars indicate SEM. (**I**) Quantification of the number of Ki67+ p75 NTR double-labeled cells express as the number of cells expressing Ki67 and p75NTR over the total number of cells (Dapi). One-way ANOVA, N=3, *p=0.0013, error bars indicate SEM. (**J**) Quantification of the number of p75NTR positive cells, expressed as the number of p75NTR positive cells over the total number of cells (Dapi). One-way ANOVA, N=3, *p=0.0001, error bars indicate SEM. (**K**) Quantification of the number of proliferating cells that also express p75NTR, expressed as the number of double-labeled Ki67+ p75 NTR cells over the total Ki67 expressing cells. One-way ANOVA, N=3, *p=0.3346, error bars indicate SEM. All the experiments were done using cells obtained from P7 rat pups.

The online version of this article includes the following source data for figure 2:

**Source data 1.** RAW Western blot for p75 and PCNA.

**Source data 2.** Quantification of p75 and PCNA for blot in *Figure 2A*.

**Source data 3.** RAW Western blot for p75 and PCNA.

**Source data 4.** Quantification of p75 and PCNA for blot in *Figure 2D*.

**Source data 5.** Quantification of Ki67/Dapi, p75+ Ki67/Dapi, p75/Dapi, and p75+ Ki67/Ki67+ cells.

*Figure 4—figure supplement 1*). Interestingly, when cells were cultured without SAG for the first 24 hr, and then with SAG for the next 24 hr, there was no significant difference in the proliferation response between the Ctrl-GFP or p75-GFP transfected cells. Moreover, there was no difference between this condition and the cells maintained in the absence of SAG for the entire 48 hr (*Figure 4A* middle panels, 4B–F, *Figure 4—figure supplement 1*). Consistently, there was a significant reduction in proliferation levels in the cells maintained without the mitogen during the first 24 hr in culture, and the re-exposure to SAG did not induce re-entry to the cell cycle (*Figure 4B, C and F*), indicating that the continued expression of p75NTR is not sufficient to maintain the CGPs in a proliferative state, thus unresponsive to the delayed SAG addition.

## P75NTR prevents CGN migration in vitro

The lack of proliferation in response to SAG observed in cells transfected with p75NTR suggests that the receptor is not involved in maintaining the cells in a proliferative state, ready to respond to a mitogen. Considering the sharp downregulation of p75NTR observed in migratory cells (*Figures 1 and 2*), an alternative explanation would be that the presence of p75NTR in proliferating GCPs prevents the onset of migration of this cell population. To assess whether the expression of p75NTR prevents granule cell migration, we used a transwell migration assay and compared the intrinsic migratory activity of dissociated granule cells obtained from P7 WT and p75NTR-/- rat pups. In this assay, the same number of cells were plated on top of the filter and the number of migrating cells toward the bottom of the filter was evaluated after 24 hr.

To evaluate the intrinsic migratory properties of each of the genotypes, no ligand was added to the transwell compartments. The p75NTR-/- cells showed a twofold increase in the number of migrating cells compared to WT cells (*Figure 5A–B*). The previous data demonstrated that mitotic GCPs maintain high levels of p75NTR, while postmitotic cells expressing neuronal differentiation markers such as DCX or βIII-tubulin completely downregulate p75NTR expression (*Figures 2 and 3*), supporting our hypothesis that p75NTR may serve to retain the proliferating progenitors in the mitogenic environment of the EGL. To test the possibility, using the transwell assay we exposed the cells to Shh (*Figure 5C* bottom panels, *Figure 5D*) to maintain them in a proliferative state expressing high levels of p75NTR, and stimulated migration using brain-derived neurotrophic factor (BDNF) (*Figure 5C* right panels, *Figure 5D*), a well-known chemoattractant for the postmitotic CGNs (*Borghesani et al., 2002*; *Zhou et al., 2007*). In the absence of Shh, BDNF stimulated the migration of CGN in agreement with the previously mentioned reports (*Figure 5C–D*), whereas, in the presence of Shh, which promotes proliferation and maintains elevated levels of p75NTR, BDNF did not induce migration (*Figure 5C–D*), indicating that only the postmitotic CGNs that have downregulated p75NTR responded to BDNF with increased migration. An alternative explanation for these findings could be that Shh maintains cells in a proliferative state, and restricts CGN migration independent of p75NTR. To test this possibility, we transfected CGNs to overexpress p75NTR and exposed the cells to BDNF in a transwell assay.

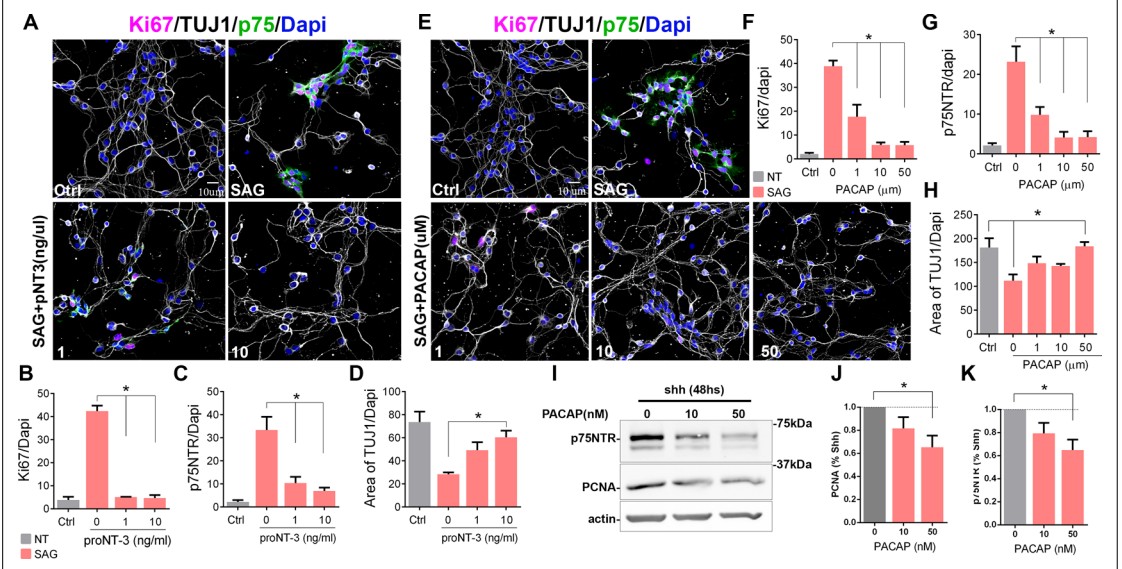

**Figure 3.** Cell cycle exit induced p75 neurotrophin receptor (p75NTR) downregulation. (**A**) Immunocytochemistry analysis of the expression of p75NTR (green), Ki67 (magenta), βIII-tubulin (white), and Dapi (blue) in granule cell cultures obtained from P7 rat pups, in response to increasing concentrations of proNT-3 after 48 hr in culture. (**B**) Quantification of the number of cells expressing Ki67 over the total number of cells (Dapi). One-way ANOVA, N=4, *p=0.0001, error bars indicate SEM. (**C**) Quantification of the number of cells expressing p75NTR over the total number of cells (Dapi). One-way ANOVA, N=4, *p=0.0001, error bars indicate SEM. (**D**) Quantification of cerebellar granule cell differentiation expressed as the total area of processes positive for TUJ1 over the total number of cells (Dapi). One-way ANOVA, N=4, *p=0.0023, error bars indicate SEM. (**E**) Immunocytochemistry analysis of the expression of p75NTR (green), Ki67 (magenta), DCX (white), and Dapi (blue) in granule cell cultures obtained from P7 rat pups in response to increasing concentrations of PACAP after 48 hr in culture. (**F**) Percentage of cells expressing Ki67 over the total number of cells (Dapi). One-way ANOVA, N=4, *p=0.0001, error bars indicate SEM. (**G**) Percentage of cells expressing p75NTR over the total number of cells (Dapi). One-way ANOVA, N=4, *p=0.0001, error bars indicate SEM. (**H**) Quantification of cerebellar granule cell differentiation expressed as the total area of processes positive for DCX over the total number of cells (Dapi). One-way ANOVA, N=4, *p=0.0114, error bars indicate SEM. (**I**) Western blot analysis of the expression of p75NTR and PCNA in granule cell cultures obtained from P7 rat pups in response to increasing concentrations of PACAP after 48 hr in culture. (**J**) Quantification of PCNA expression in cerebellar granule neuron (CGN) cultures expose to PACAP and normalize to Shh alone. One-way ANOVA, N=4, *p=0.0423, error bars indicate SEM. (**K**) Quantification of p75NTR expression in CGN cultures expose to PACAP and normalize to Shh alone. One-way ANOVA, N=4, *p=0.0251, error bars indicate SEM. Quantification of the area of TUJ1 processes was done using a custom-built ImageJ macro. All the experiments were done using cells obtained from P7 rat pups.

The online version of this article includes the following source data, source code, and figure supplement(s) for figure 3:

**Source code 1.** ImageJ macro to count TUJ1 in the processes.

**Figure supplement 1.** Cell cycle exit induced p75 neurotrophin receptor (p75NTR) downregulation.

**Source data 1.** Quantification of p75, TUJ1, and Ki67 after proNT-3 incorporation.

**Source data 2.** Quantification of p75, TUJ1, and Ki67 after PACAP incorporation.

**Source data 3.** Raw Western blot for p75 and PCNA after PACAP incorporation.

**Source data 4.** WB (Western blot) quantification of p75 and PCNA after PACAP incorporation.

Our results showed an increase in migration in response to BDNF in the CGN transfected with a control GFP construct (*Figure 5E* top panel, *Figure 5F*), similar to the CGN response to BDNF in WT non-transfected cells (*Figure 5C–D*). However, the chemotactic effect of BDNF was lost in the CGN overexpressing p75NTR (*Figure 5E* bottom panel, *Figure 5F*), suggesting that p75NTR is sufficient to prevent CGN migration even in the presence of a chemotactic signal such as BDNF.

## P75NTR expression prevents CGN migration in cerebellar slices

To directly evaluate the effects of p75NTR expression during CGN migration in cerebellar tissue, we transfected p75NTR into the EGL in cerebellar organotypic slices and tracked migrating CGNs using time-lapse microscopy. P7 rat cerebellum slices were electroporated either with a Ctrl-GFP or p75NTR-GFP construct, specifically into the EGL. In slices transfected with the Ctrl-GFP vector, we observed cells migrating toward the IGL (*Figure 5G* top panels, *Figure 5—video 1*). The migration

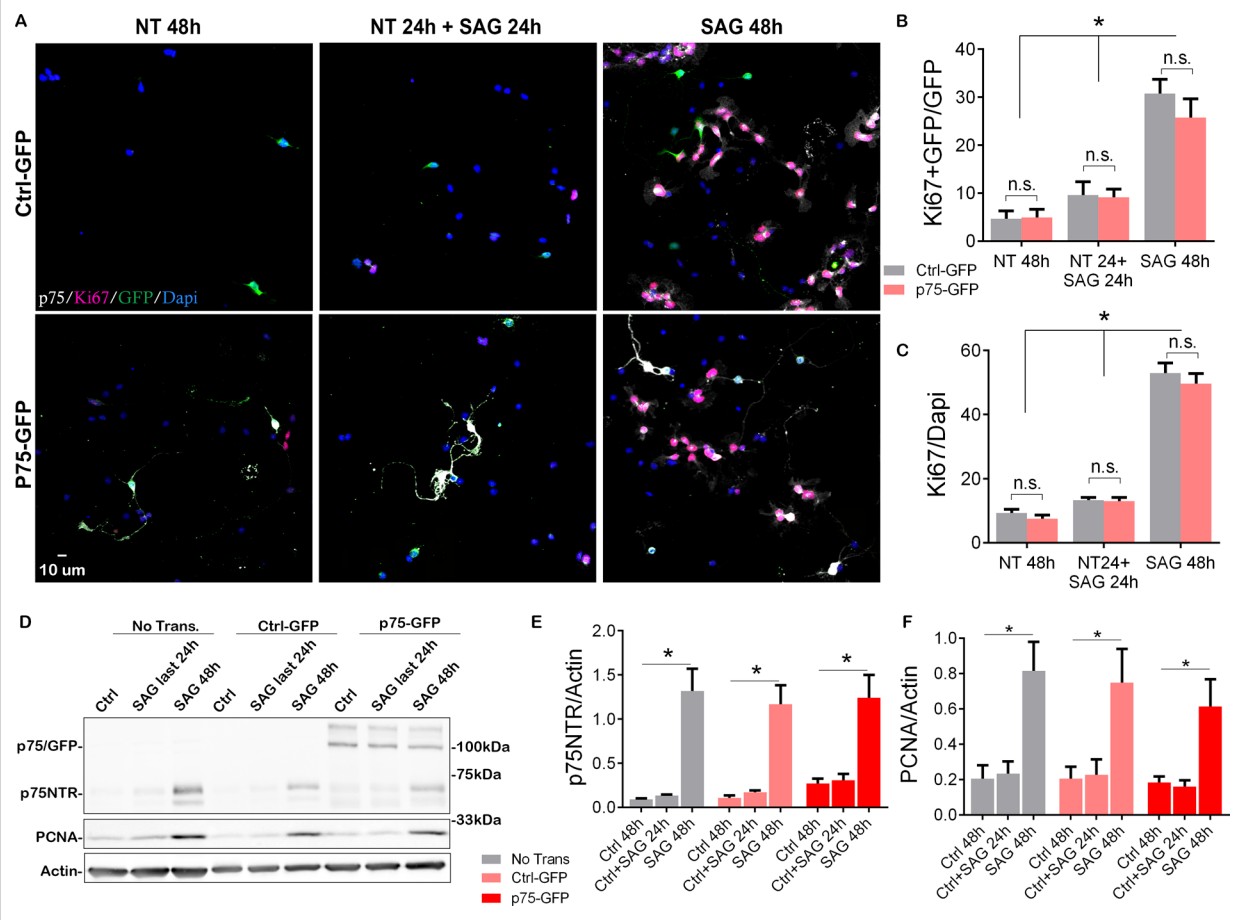

**Figure 4.** Overexpression of p75 neurotrophin receptor (p75NTR) is not sufficient to maintain granule cells in a proliferative state. (**A**) Immunostaining analysis of the expression of p75NTR (white), GFP (green), Ki67 (magenta), and Dapi (blue) in granule cell cultures obtained from P7 rat pups transfected with Ctrl-GFP (top row) or p75-GFP (bottom row) construct. Left panels, cells were maintained in the absence of mitogen for 48 hr. Right panels, transfected cells were maintained in SAG for 48 hr. Middle panel cells were maintained without SAG for the first 24 hr in culture, and in the presence of the mitogen only in the last 24 hr in culture. (**B**) Quantification of the transfected cells that are proliferating, expressed as the double-labeled Ki67/GFP over the total number of transfected cells (GFP+ cells). Two-way ANOVA, *p=0.0001, N=4, error bars indicate SEM. (**C**) Quantification of the total number of proliferating cells expressed as the percentage of Ki67+ cells over the total number of cells (Dapi). Two-way ANOVA, N=4, *p=0.0001, error bars indicate SEM. (**D**) Western blot analysis of the expression levels of p75NTR and PCNA in granule cell cultures obtained from P7 rat pups, first three lanes no transfected cells, last six lanes' cells transfected with Ctrl-GFP or p75-GFP. (**E**) Quantification of the expression levels of endogenous p75NTR, Two-way ANOVA, N=4, *p=0.0001, error bars indicate SEM. (**F**) Quantification of the expression levels of PCNA, Two-way ANOVA, N=4, *p=0.0001, error bars indicate SEM. All the experiments were done using cells obtained from P7 rat pups.

The online version of this article includes the following source data and figure supplement(s) for figure 4:

**Figure supplement 1.** Overexpression of p75 neurotrophin receptor (p75NTR) in granule cell precursors (GCPs).

**Source data 1.** Quantification of Ki67+ GFP/GFP and Ki67/Dapi cells.

**Source data 2.** Raw Western blot for p75 and PCNA after p75-GFP transfection.

**Source data 3.** Western Blot quantification of p75 and PCNA after p75-GFP transfection.

parameters, speed and distance, were consistent with previously reported values (*Figure 5H–I*; *Yacubova and Komuro, 2002*). These cells displayed the classic morphology of migrating neurons, a leading process extended toward the IGL was observed and the soma showed a 'saltatory' migration pattern, that is, alternating stationary and displacement periods of the cell body (*Yacubova and Komuro, 2002*). However, when p75NTR was overexpressed, the transfected neurons remained in the EGL. In contrast to the control conditions, these cells maintained a rounded morphology and never extended a process in any direction, resembling a non-migrating neuron (*Figure 5G* bottom panels, *Figure 5—video 2*). These results strongly support the possibility that the presence of p75NTR is

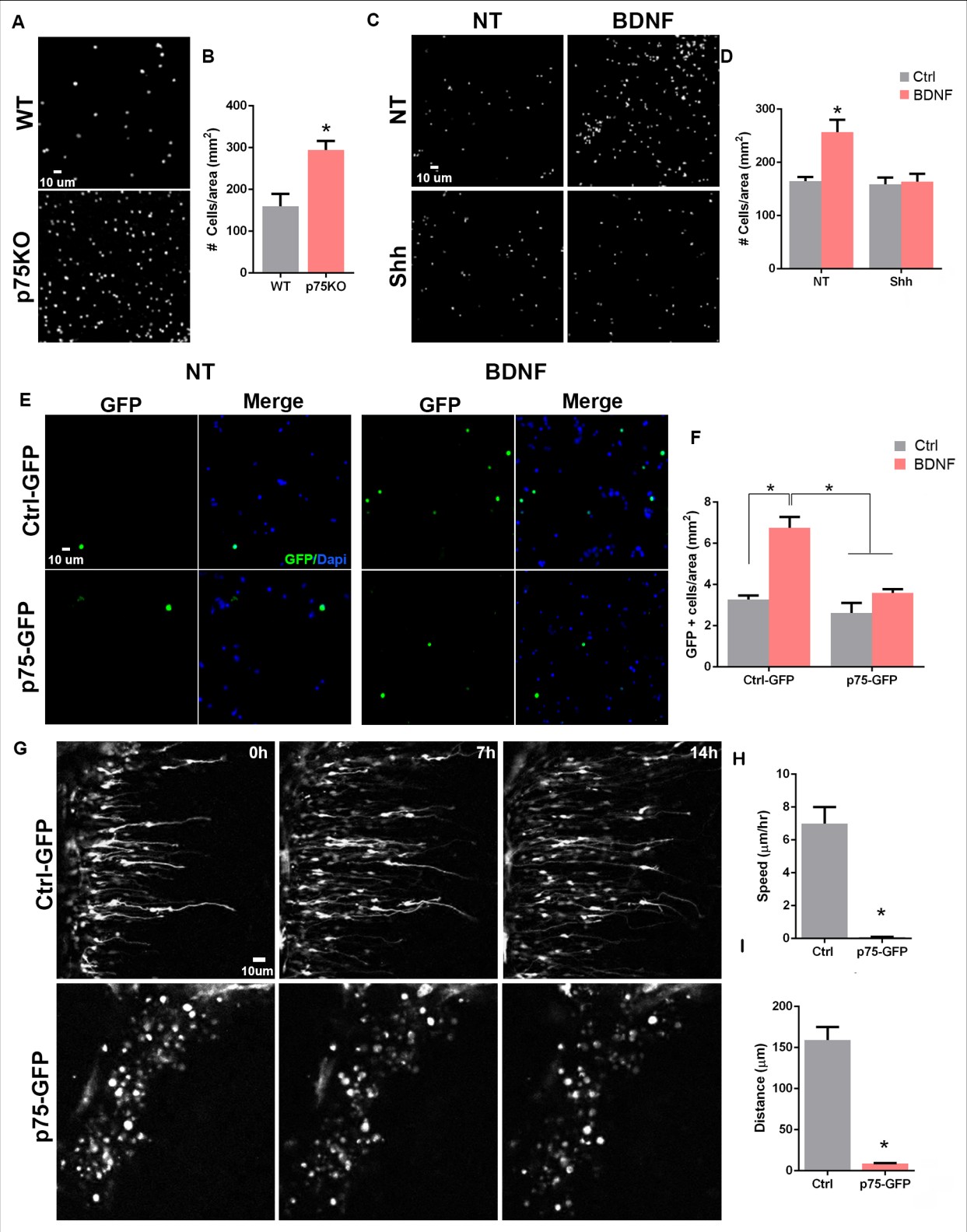

**Figure 5.** p75 neurotrophin receptor (p75NTR) prevents cerebellar granule neuron (CGN) migration in vitro. (**A**) Migration analysis using transwell assay. Dapi staining of cells obtained from WT or p75NTR-/- P7 rat pups. (**B**) Quantification of CGN migration expressed as the density of Dapi+ cells at the bottom of the filter after 24 hr. At the beginning of the experiment the same number of cells were plated on top of the filter. No ligand was added to stimulate the migration. Unpaired t-test, N=4, *p=0.01, error bars indicate SEM. (**C**) Migration analysis using transwell assay in granule cells obtained from P7 rat pups exposed to Shh in the top and bottom compartment and brain-derived neurotrophic factor (BDNF) in the bottom compartment.

*Figure 5 continued on next page*

*Figure 5 continued*

(**D**) Quantification of CGN migration expressed as the density of Dapi+ cells at the bottom of the filter after 24 h of BDNF exposure. Two-way ANOVA, N = 4, p = 0.0186, error bars indicate SEM. (**E**) Migration analysis using transwell assay in granule cells obtained from P7 rat pups transfected with Ctrl-GFP or p75-GFP construct and expose to BDNF in the bottom compartment. GFP immunostaining of the transfected cells. (**F**) Quantification of CGN migration expressed as the density of transfected GFP+ cells at the bottom of the filter after 24 hr of BDNF exposure. Two-way ANOVA, N=4, *p=0.0065, error bars indicate SEM. (**G**) Time-lapse pictures from cerebellar organotypic slices were obtained from P7 rat pups and transfected with Ctrl-GFP (top panels) or p75NTR-GFP (bottom panels). (**H**) Mean migration speed is expressed as the total distance migrated over the total time of the experiment. Unpaired t-test, N=3, p = 0.0001, error bars indicate SEM. (**I**) Total distance migrated is expressed as the mean distance migrated per cell. Unpaired t-test, N=3, p = 0.0003, error bars indicate SEM. The experiments were done using cells from P7 rat pups.

The online version of this article includes the following video and source data for figure 5:

**Figure 5—video 1.** Timelapse image of P7 rat pup cerebellum transfected with Ctrl-GFP.
https://elifesciences.org/articles/79934/figures#fig5video1

**Figure 5—video 2.** Timelapse image of P7 rat pup cerebellum transfected with p75-GFP.
https://elifesciences.org/articles/79934/figures#fig5video2

**Source data 1.** Quantification of Dapi cells/area of filter WT vs. p75KO.

**Source data 2.** Quantification of Dapi cells/area of filter expose to brain-derived neurotrophic factor (BDNF) and/or Shh.

**Source data 3.** Quantification of GFP+ cells/ area of filter after p75-GFP transfection.

sufficient to prevent granule cell migration, and the receptor needs to be downregulated before the cells start to migrate.

## The absence of p75NTR in GCPs increases granule cell migration in vivo

During cerebellar development, p75NTR is expressed in GCPs and Purkinje cells (see *Figure 1*). To analyze the specific role of p75NTR in the transition of GCPs from a proliferating to a migrating population in vivo, we mated *p75*^FL/FL^ mice (*Bogenmann et al., 2011*) with *Atoh1*^Cre^ mice (Jackson Laboratory). In the cerebellum, these mice lack p75NTR expression specifically in the EGL while maintaining WT expression of the receptor in the Purkinje cells (*Zanin et al., 2016*). We injected EdU into P7 mice and waited for 24 or 48 hr before perfusing the animals to allow the dividing cells in the EGL to incorporate EdU and eventually exit the cell cycle and migrate toward the IGL. Cells positive for EdU, located in the IGL, are postmitotic GCPs that migrated in the time between the EdU injection and euthanizing the animal. At both time points, 24 and 48 hr, a significantly increased number of EdU+ cells were found in the IGL of the *p75*^FL/FL^; *Atoh1*^Cre^ mice compared to WT animals (*Figure 6A–D*). Developmental differences between anterior and posterior lobes have been previously reported (*Martinez et al., 2013*), however, we observed a significant increase of EdU+ cells in the anterior lobes (lobule 5) and the posterior lobes (lobules 6 and 9) of *p75*^FL/FL^; *Atoh1*^Cre^ (*Figure 6C–D*). PAX6 staining confirmed the cerebellar granule neuron identity of the EdU+ cells in the IGL (*Figure 6E*). Cells positive for EdU in the white matter were negative for PAX6, therefore excluded from the analysis (*Figure 6E*, dotted line).

The increased presence of granule cells in the IGL observed in the *p75*^FL/FL^; *Atoh1*^Cre^ mice might be due to abnormal early migration of the GCPs if the cells started the migratory process before they exited the cell cycle. To assess whether cells migrated inappropriately while still in the cell cycle, we stained the EdU-injected animals with Ki67, a proliferation marker present in actively proliferating cells during the entire cell cycle (*Gerdes et al., 1983*; *Scholzen and Gerdes, 2000*). A double-labeled cell (EdU/Ki67 positive cell) in the IGL would indicate premature GCP migration while the cells were still proliferating. Our results showed that in WT as well as in *p75*^FL/FL^; *Atoh1*^Cre^ animals, the number of EdU/Ki67 positive cells present in the IGL was close to zero, and no difference was observed between the genotypes (*Figure 6F–G*), indicating that cell migration of proliferating GCPs was not observed when we removed p75NTR, therefore cells still exited the cell cycle before migration.

## RhoA inhibits GCP radial migration

RhoA is a member of the Rho GTPase family of proteins which play an important role in cell migration via cytoskeletal regulation (*Ridley and Hall, 1992*; *Ridley, 1995*). RhoA activity is known to be regulated by p75NTR (*Yamashita et al., 1999*; *Yamashita and Tohyama, 2003*), and granule cells from p75NTRKO mice show reduced levels of active RhoA (*Figure 7A*). Therefore, to test the involvement

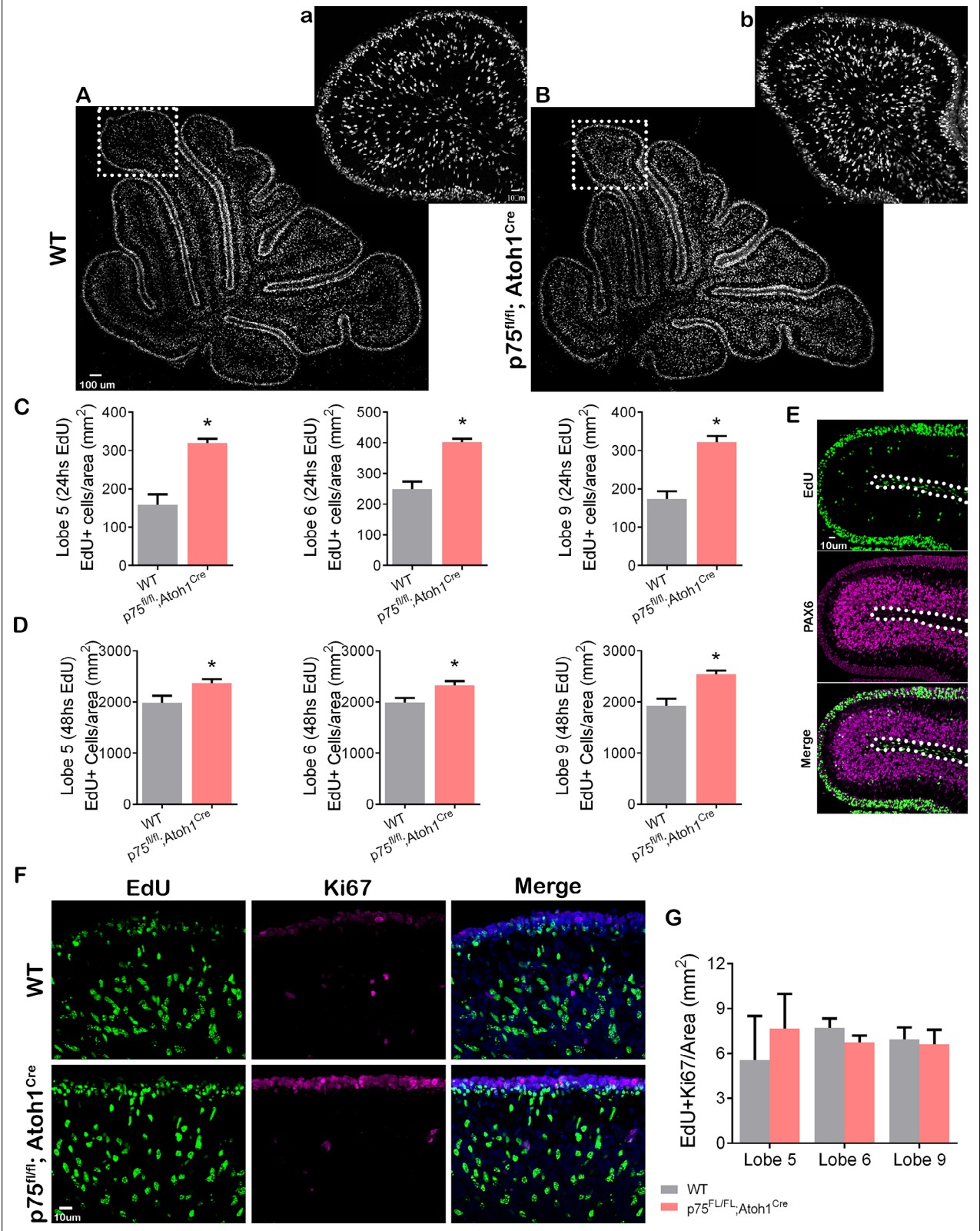

**Figure 6.** p75 neurotrophin receptor (p75NTR) prevents cerebellar granule neuron (CGN) migration in vivo. (**A–B**) Immunohistochemistry for EdU of cerebellar sections from P9 mouse. (**a–b**) High magnification of the inset shown in panels (**A–B**). (**C–D**) Quantification of the density of migrating cells in the cerebellum of mouse pups. Migrating cells are expressed as the total number of EdU positive cells in the internal granule layer (IGL) per mm². (**C**) P8 mouse pups injected with EdU 24 hr before euthanizing the animal. Unpaired t-test, WT N=6, *p75$^{fl/fl}$; Atoh1$^{Cre}$* N=4, Lobe 5 *p=0.0001, Lobe 6 *p=0.0001,

*Figure 6 continued on next page*

*Figure 6 continued*

Lobe 9 *p=0.0001, error bars indicate SEM. (**D**) P9 mouse pups were injected with EdU 48 hr before euthanizing the animal. Unpaired t-test, WT N=9, *p75*$^{fl/fl}$; *Atoh1*$^{Cre}$ N=9, Lobe 5 *p=0.0302, Lobe 6 *p=0.0117, Lobe 9 *p=0.0012, error bars indicate SEM. (**E**) Cerebellar section from a WT P8 mouse injected with EdU 24 hr before euthanizing the animal. EdU (green) and Pax6 (granule cell marker, magenta). (**F**) Immunohistochemistry of cerebellar sections from P9 mouse pups injected with EdU 48 hr before euthanizing the animal. EdU (green), Ki67 (proliferation marker, magenta), and Dapi (blue). (**G**) Quantification of the migrating cells that continued to express Ki67, expressed as the density of EdU/Ki67 double-labeled cells in the IGL. One-way ANOVA, N=3, p=0.2516, error bars indicate SEM. The experiments presented here were done using P7 to P9 mouse pups.

The online version of this article includes the following source data for figure 6:

**Source data 1.** Quantification of EdU+ cells/area after 24 hr of EdU injection.

**Source data 2.** Quantification of EdU+ cells/area after 48 hr of EdU injection.

**Source data 3.** Quantification of EdU+ and Ki67+ cells/area in the internal granule layer (IGL).

of RhoA in CGN migration, we used the Rho kinase inhibitor, Y27632, to block the RhoA signaling pathway and analyzed CGN migration using the transwell assay. Even in the absence of a chemoattractant, inhibition of the RhoA pathway significantly increased the number of migrating cells compared to the control condition (*Figure 7B*, black bars), similar to the p75NTR knockout CGNs. Moreover, the inhibition of Rho kinase did not affect the ability of BDNF to promote migration, suggesting that RhoA is not required for CGN migration (*Figure 7B*, gray bars). Since inhibition of RhoA signaling promoted migration, we assessed whether RhoA must be downregulated for CGN migration to occur. Organotypic cultures transfected with Ctrl-GFP migrated normally when exposed to a Rho kinase inhibitor (*Figure 7C* and *Figure 7—video 1*), confirming that RhoA activity is not necessary for migration to occur (compare *Figure 5—video 1* and *Figure 7—video 1*). However, overexpression of a constitutively active form of RhoA (CA-RhoA) in the EGL of P7 cerebellar slices inhibited migration toward the IGL (*Figure 7D* and *Figure 7—video 2*), similar to the overexpression of p75NTR-GFP (see *Figure 5E*), and the majority of the cells maintained a rounded shape with no extended process in the transfected cells (*Figure 7D* and *Figure 7—video 2*). After finishing the time-lapse video, the cerebellar slices were fixed and stained for DCX, to identify the migrating cells, and GFP to identify the transfected cells overexpressing the CA-RhoA (*Figure 7E*). The cells expressing the CA-RhoA remained in the EGL, and no GFP was detected in the IGL. Consistent with the anti-migratory role of RhoA, no colocalization between the CA-RhoA cells and DCX was observed (*Figure 7E*).

## P75NTR inhibits GCP migration via RhoA activation

p75NTR can interact with Rho-GDI at the plasma membrane of GCPs, helping to maintain high levels of active RhoA in a ligand-independent fashion (*Yamashita et al., 1999*; *Yamashita and Tohyama, 2003*). We have previously demonstrated that active RhoA was necessary for the GCPs to remain in the cell cycle, and inhibition of RhoA elicited cell cycle exit of GCPs, which was correlated with a downregulation of p75NTR (*Zanin et al., 2019*). Furthermore, GCPs obtained from p75KO rats showed a significant reduction in the levels of active RhoA compared to WT cells (*Figure 7A*). These results suggest that the elevated levels of p75NTR present in the proliferating GCPs might prevent cell migration via RhoA activation and retain the GCPs in the mitogenic environment of the EGL. To address this possibility, we transfected cerebellar slices with a p75-GFP construct and added the RhoA inhibitor Y27632 directly into the bath. Our results showed that inhibiting RhoA was sufficient to induce migration of the CGN (*Figure 7F* and *Figure 7—video 3*). These results confirm that the inhibition of CGN migration induced by p75NTR requires the activation of RhoA.

## Discussion

Neuronal migration is one of the fundamental mechanisms by which the nervous system acquires its final shape and function. Defects in neuronal migration often result in neuronal cell death or misplacement of a neuronal population, leading to a variety of disorders including epilepsy, mental retardation, and cortical dysplasia, among other neurological disorders (*Copp and Harding, 1999*; *Pilz et al., 2002*; *Moffat et al., 2015*). Although there is considerable emphasis on understanding the mechanisms that promote neuronal migration, as well as identifying potential signals that support cell guidance, little is known about the mechanisms that restrict migration and regulate the timing of the onset of migration. The balance between promoting and preventing migration is critical for the

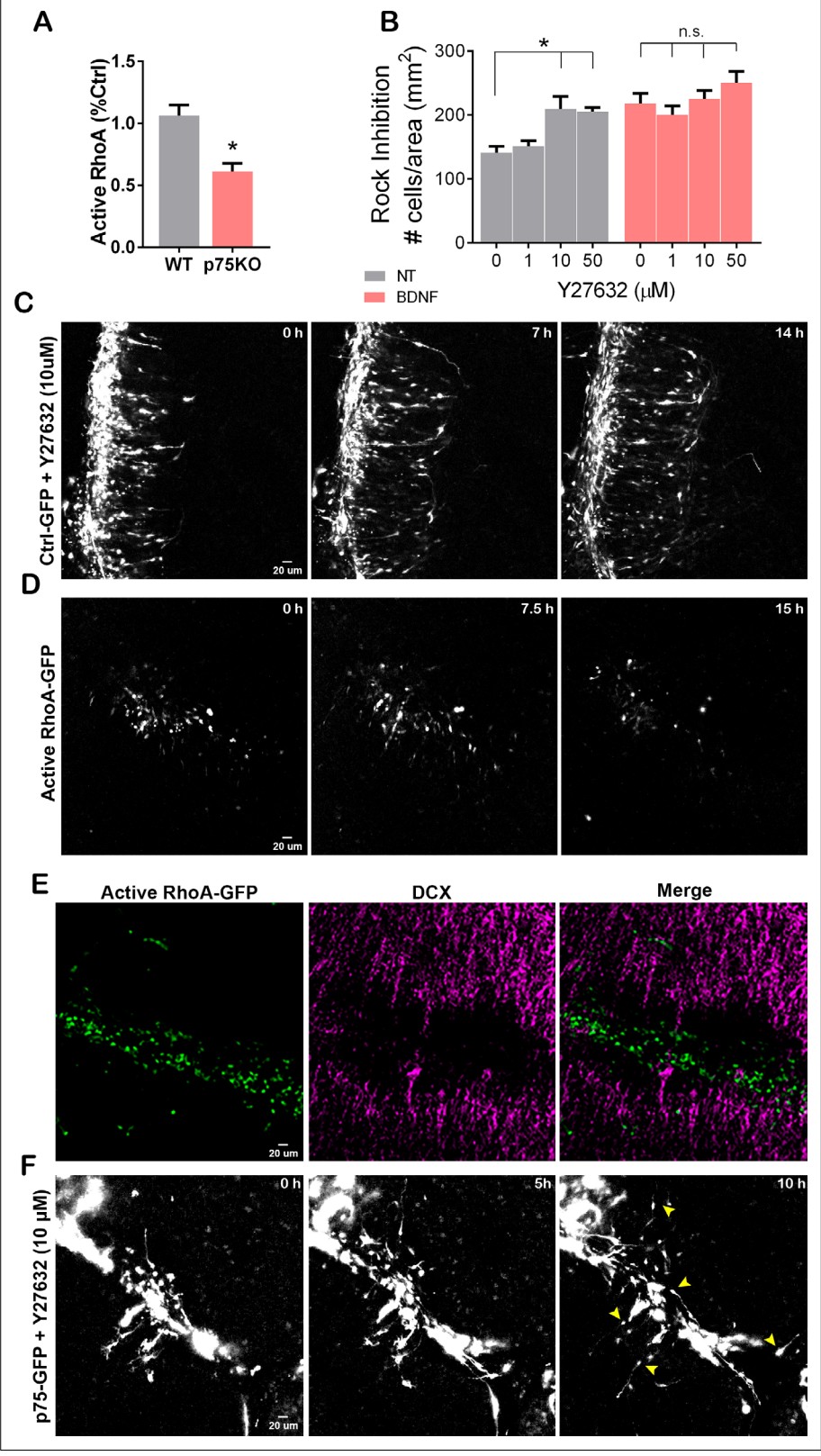

**Figure 7.** RhoA activation prevents cerebellar granule neuron (CGN) migration. (**A**) Quantification of the levels of active RhoA in granule cell cultures from WT or p75KO P7 rat pups after 48 hr in culture. Unpaired t-test, N=4, *p=0.0052, error bars indicate SEM. (**B**) Migration analysis using transwell assay in cells exposed the Rock inhibitor, Y27632 (top and bottom compartment), and brain-derived neurotrophic factor (BDNF) (bottom compartment).

*Figure 7 continued on next page*

*Figure 7 continued*

Two-way ANOVA, N=5, *p=0.0005 error bars indicate SEM. (**C**) Time-lapse pictures from cerebellar organotypic slices from P7 rat pups, transfected with Ctrl-GFP in the presence of the Rock inhibitor, Y27632. (**D**) Time-lapse pictures from cerebellar organotypic slices from P7 rat pups, transfected with a constitutively active RhoA construct. (**E**) Immunostaining of the organotypic slice shown in D after clearing the section using the iDisco method, RhoA-GFP (green), and DCX (magenta). (**F**) Time-lapse pictures from cerebellar organotypic slices from P7 rat pups, transfected with a p75NTR-GFP construct and exposed to Rock inhibitor (Y27632). Arrowheads indicate the migrating neurons where a leading process is observed. The experiments presented here were done using slices obtained from P7 rat pups.

The online version of this article includes the following video and source data for figure 7:

**Source data 1.** Quantification of active RhoA (G-Lisa) WT vs p75KO.

**Source data 2.** Quantification of Dapi+ cells/area of filter expose to Rock Inhibitor.

**Figure 7—video 1.** Time-lapse image of P7 rat pup cerebellum transfected with Ctrl-GFP, in the presence of Rock inhibitor (Y27632 10 μm).

https://elifesciences.org/articles/79934/figures#fig7video1

**Figure 7—video 2.** Time-lapse image of P7 rat pup cerebellum transfected with a constitutive active from of RhoA-GFP.

https://elifesciences.org/articles/79934/figures#fig7video2

**Figure 7—video 3.** Time-lapse image of P7 rat pup cerebellum transfected with p75-GFP in the presence of the Rock inhibitor (Y27632 10 μm).

https://elifesciences.org/articles/79934/figures#fig7video3

timely coordination of the developing nervous system since it positions the correct neuronal types at a precise time for the establishment of functional neuronal connections. In the cerebellum, it has been proposed that the EGL represents a mitogenic niche permissive for granule cell proliferation, and migration away from this niche results in granule cell differentiation (*Choi et al., 2005*). Consistent with this hypothesis, in the present work, we identify a novel function for p75NTR in restricting the initiation of migration of cerebellar GCPs, providing a mechanism that explains how GCPs are retained in the EGL.

## P75NTR is necessary to maintain CGPs undifferentiated but is not sufficient to keep them in a proliferative state

P75NTR is highly expressed during development in both neuronal and non-neuronal precursors. Embryonic expression of P75NTR starts as early as the blastocyst stage and continues throughout development (*Moscatelli et al., 2009*; *Vandamme and Berx, 2019*). In addition, p75NTR is expressed in a wide variety of adult stem cells, such as the subventricular zone (*Giuliani et al., 2004*; *Petratos et al., 2004*) and the hair follicle (*Botchkareva et al., 1999*; *Botchkarev et al., 2000*) although its function in these cell populations is not clear. Due to the early expression pattern and the presence of the receptor in embryonic and adult stem cell populations, it has been suggested that p75NTR might contribute to maintaining cells in an undifferentiated state. This hypothesis is further supported by findings in the cancer field where p75NTR regulates key features of cancer cells such as stem cell-like phenotypes and cell migration (*Vidal and Redmer, 2020*). Our results further support this possibility since the morphogen Shh induces (or at least maintains) the expression of p75NTR in GCPs. Shh is secreted by Purkinje cells and is a well-established mitogen for GCPs in the EGL during early cerebellar development. p75NTR is expressed in all the proliferating GCPs in the external EGL and is downregulated in postmitotic CGNs. The same expression pattern is observed in vitro, where in the presence of Shh the cells continued to proliferate and express p75NTR. Moreover, stimulation of GCP cell cycle exit using proNT-3 or PACAP induced a reduction in p75NTR expression together with the cessation of proliferation and increased differentiation. ProNT-3 and PACAP both induced cell cycle exit even in the presence of Shh, suggesting that GCP cell cycle exit is an active mechanism depending on the proper signals and not just the absence of the mitogen. Altogether, our results suggest a potential role of p75NTR in preventing cell differentiation, and it might contribute to maintaining a pool of undifferentiated granule cells during cerebellar development.

In our previous work we showed that p75NTR regulates the timely exit of the cell cycle in GCPs by controlling the speed of proliferation, and since p75NTR is highly expressed in all the proliferating GCPs, an open question still is whether the expression of this receptor maintains the cells in a permissive state for proliferation. In this study, we confirmed that continued expression of p75NTR was not sufficient to maintain the GCPs in a proliferative state, once the mitogen was absent for at least 24 hr and the cells had begun to differentiate, since re-addition of Shh did not evoke cell cycle re-entry in cells that continued to express p75NTR. Although this result argues against the possible role of p75NTR in maintaining cells in an undifferentiated state, these data must be interpreted with caution since Shh also activates other signaling pathways that might act alone or in concert with other factors to maintain cells in a proliferative state. Thus, removing Shh for 24 hr might induce the activation (or inhibition) of other signaling pathways also required to maintain cells in the cell cycle. Further experiments are needed to elucidate these potential pathways, and more important, whether they interact with the neurotrophin signaling to maintain cells undifferentiated.

## P75NTR prevents granule cell migration

The well-defined boundary between the migrating granule cells expressing DCX and the proliferating GCPs expressing p75NTR suggested that the receptor might be involved in regulating the onset of GCP migration. In this study, we demonstrated that the presence of p75NTR was sufficient to prevent the migration of GCPs. Using a transwell assay we observed that cells obtained from p75KO rats have an intrinsically higher rate of migration compared to WT cells. Moreover, cells that were grown in the presence of Shh, which induces proliferation of the GCPs while maintaining elevated expression levels of p75NTR, were sufficient to reduce the migration of GCPs, even in the presence of BDNF, a known chemotactic signal for these cells. Although maintaining the cells in the cell cycle may prevent the onset of migration by a variety of mechanisms, we demonstrated that the overexpression of p75NTR in CGNs was sufficient to block migration even in the presence of BDNF. These observations raise the possibility that the high expression of p75NTR in the presence of Shh serves to keep the cells in the proliferative environment of the EGL.

To further test this possibility, we used time-lapse microscopy in cerebellar organotypic cultures, maintaining the cells in the proper tissue environment and in the presence of the molecular signals that regulate GCP proliferation and migration. Overexpression of p75NTR was sufficient to block GCP migration from the EGL. Interestingly, these cells maintained a rounded shape and no process was observed in any direction, unlike the cells in the control conditions where a clear leading process was observed in the cell edge facing the IGL. Since the elaboration of the leading process is critical for the onset of GCP migration, the continued expression of p75NTR may prevent the changes in cytoskeletal dynamics required for migration to occur. Although our results suggest that p75NTR might act independent of a ligand, we cannot completely rule out the autocrine or paracrine release of a ligand that blocks GCP migration. Granule cells can synthesize BDNF, and its precursor form proBDNF has been proposed as a potential anti-migratory signal for GCP (*Xu et al., 2011*). It has been reported that proBDNF acting via p75NTR is a negative regulator of GCP migration. Surprisingly the authors showed using p75KO animals that the absence of the receptor significantly reduces GCP migration, which is not expected if proBDNF/p75NTR signaling is anti-migratory.

During cerebellar development, p75NTR is expressed in the GCPs throughout the entire EGL surface, including every lobe of the vermis and the hemispheres (*Rahimi Balaei et al., 2016*; *Carter et al., 2003*; *Zanin et al., 2016*). Our in vivo experiments demonstrated that the absence of p75NTR was sufficient to increase GCP migration in both the anterior and posterior lobes, suggesting a common mechanism regulating the onset of migration throughout the entire cerebellum. These findings are particularly interesting since different areas and lobes of the cerebellum are associated with different behaviors (*Voogd and Glickstein, 1998*; *Buckner et al., 2011*; *Buckner, 2013*). Since the absence of the receptor affected the entire cerebellum, our findings reveal an important role for p75NTR, dysregulated migration in the absence of this receptor can potentially impact a wide range of behaviors.

## Rho A is required to prevent GCP migration

RhoA has been associated both positively and negatively with cell migration via cytoskeletal regulation in different cell populations including cerebellar granule cells (*Bito et al., 2000*; *Fukata et al.,*

*2003*; *Amano et al., 2010*) and Schwann cells (*Yamauchi et al., 2004*). In CGNs, p75NTR can constitutively maintain elevated levels of active RhoA in the absence of ligand (*Yamashita et al., 1999*; *Yamashita and Tohyama, 2003*). In our previous study, we demonstrated that proNT-3 induced cell cycle exit of the CGPs (*Zanin et al., 2016*) and at the same time it induced a reduction in the levels of active RhoA (*Zanin et al., 2019*). Consistent with these results, we showed that p75NTR-/- cells have a reduced level of active RhoA compared to WT cells, and inhibition of RhoA was sufficient to promote the migration of GCPs. Moreover, overexpressing a constitutively active form of RhoA prevented the migration of GCPs, while blocking RhoA activity in organotypic cultures did not affect GCP migration. Taking into consideration our previous and current findings, the inhibition of migration induced by p75NTR might be due to the maintenance of active RhoA observed in the presence of the receptor. Upon cell differentiation, the levels of p75NTR are reduced inducing a reduction in the activity of RhoA, allowing the cells to migrate.

In summary, our findings demonstrate an anti-migratory effect of p75NTR in GCPs. Although we do not discard the involvement of an endogenously produced neurotrophin ligand to regulate migration, our results suggest that an un-liganded p75NTR is responsible for preventing CGN migration. Our discovery emphasizes the multiple roles of p75NTR, depending on the nature of the ligand, the receptor complex involved, and now the possibility of the activation of intracellular signals in a ligand-independent manner. All these possibilities highlight the complexity of p75NTR signaling and the urgent necessity to address them in a cell-specific context, as they could provide an understanding of normal embryonic development and adult brain function while providing new targets for potential novel therapeutics.

# Materials and methods

## Animals used

All animal studies were conducted using the National Institutes of Health guidelines for the ethical treatment of animals with the approval of the Rutgers Animal Care and Facilities Committee.

*P75^FL/FL*; *Atoh1^Cre* mice were generated by crossing mice with homozygous floxed alleles of *p75NTR* (*p75^FL/FL*) (*Bogenmann et al., 2011*) with *Atoh1^Cre* mice (Jackson Laboratory, Cat# 011104. RRID: IMSR_JAX:011104) until homozygotes *p75^FL/FL* and hemizygotes *Atoh1^Cre* were obtained (*p75^FL/FL*; *Atoh^-Cre+/-*). Pups obtained by crossing these animals were obtained postnatally at different developmental ages. Both males and females were included in the analysis. The genotype of *p75NTR^FL/FL*; *Atoh1^Cre* animals was confirmed by PCR, and the absence of p75NTR in the EGL was confirmed by immunostaining.

P7 rat pups lacking p75NTR were obtained from SAGE/Horizon Laboratories, CAT# HsdSage:SD-*Ngfr^em1Sage*, and confirmed by us using PCR, Western blot, and immunostaining. P7 wild type rat pups were purchased from Charles River, CAT# 001CD. Both males and females were included in the analysis.

## Primary cerebellum cell cultures

All animal studies were conducted using the National Institutes of Health guidelines for the ethical treatment of animals with the approval of the Rutgers Animal Care and Facilities Committee. Cerebella were removed under sterile conditions from P7 pups after euthanizing them with $CO_2$. Meninges and small blood vessels were removed under a dissecting microscope. The tissue was minced and dissociated using the papain dissociation kit (Worthington LK003150). Dissociated neurons were plated onto 24-well plates ($1\times10^5$ cells in 300 µl of serum-free media [SFM]) or 6-well plates ($1.5\times10^6$ cells per well in 1 ml of SFM) coated with poly-D-lysine (0.1 mg/ml). SFM consisted of 1:1 MEM and F12, with glucose (6 mg/ml), insulin (2.5 mg/ml), putrescine (60 µM), progesterone (20 nM), transferrin (100 µg/ml), selenium (30 nM), penicillin (0.5 U/ml), and streptomycin (0.5 µg/ml).

## Immunohistochemistry

Animals were deeply anesthetized with ketamine/xylazine and perfused with 4% PFA/PBS. Brains were removed and postfixed in 4% PFA/PBS overnight at 4°C, then cryopreserved with 30% sucrose. Sections (12 µm) were cut using a Leica cryostat and mounted onto charged slides. Sections were permeabilized with 0.5% Triton in PBS for 20 min and blocked with 1% BSA and 5% donkey serum

in PBS for 1 hr at room temperature (RT). EdU stained was developed following the manufacturer's instructions (Thermo Fisher Scientific Cat# C10337). Antibody staining was started immediately after finishing EdU staining. Primary and secondary antibodies were prepared in 1% BSA. Sections were incubated with primary antibodies overnight at 4°C in a humidified chamber. Antibodies used were: Ki67 (Abcam 15580, RRID: AB_443209, 1:500), anti-p75 (R&D AF367, RRID: AB_2152638, 1:500), anti-p75 (Millipore MAB365, RRID: AB_2152788, 1:1000), anti-PAX-6 (BD Bioscience, RRID: AB_397991, 1/500), anti-DCX (Abcam 18723, RRID: AB_732011, 1/500), and anti-TAG1 (R&D Systems AF1714, RRID: AB_2245173, 1/500). All secondary antibodies were diluted at 1:1000 and incubated for 1 hr at RT. Sections were mounted using DAPI Fluoromount-G (Southern Biotech #0100-20). Controls for immunostaining included incubation with secondary antibodies in the absence of primary antibodies.

## Immunocytochemistry

Cells were fixed in 4% PFA/PBS for 20 min at RT and permeabilized with 0.5% Triton in PBS for 15 min and blocked with 1% BSA and 5% donkey serum in PBS for 1 hr at RT. EdU stained was developed following the manufacturer's instructions (Thermo Fisher Scientific Cat# C10337). Antibody staining was started immediately after finishing EdU staining. Primary and secondary antibodies were prepared in 1% BSA. Cells were incubated with primary antibodies overnight at 4°C in a humidified chamber. Antibodies used were: anti-p75 (R&D AF367, RRID: AB_2152638, 1:500), anti-DCX (Abcam, RRID: AB_732011, 1/500), Ki67 (Abcam 15580, RRID: AB_443209, 1:500), TUJ1 (Promega, RRID: AB_430874, 1/1000), anti-GFP (Sigma, RRID: AB_259941, 1/500). All secondary antibodies were diluted at 1:1000 and incubated for 1 hr at RT. Sections were mounted using DAPI Fluoromount-G (Southern Biotech #0100-20). Controls for immunostaining included incubation with secondary antibodies in the absence of primary antibodies. Cells were quantified using ImageJ version 1.51s. The quantification of TUJ1 in the process was performed using an ImageJ custom-built macro (*Figure 3—source code 1*).

## Cell culture transfection

P7 dissociated granule cells were obtained as described above and transfected using Nucleofector II (Lonza) following the manufacturer's specifications. Briefly, $2–4 \times 10^6$ cells were spun down at 80× *g* for 10 min. The cell pellet was resuspended in 100 µl of transfection buffer (Lonza). The corresponding DNA construct to be transfected was added and mixed in the resuspended cells at a concentration of 1 µg DNA/million cells. Immediately after mixing, the cells were electroporated using a Nucleofector II (Lonza), using the G-013 program. After electroporation, 500 µl of SFM supplemented with 1% B27 was added to the cells and transferred to a 15 ml conical tube. The cell suspension was left to recover for 5 min in the incubator (37°C, 5% $CO_2$). The cells were plated at a concentration of $1\times10^5$ per well in a 24-well plate with poly-D-lysine-coated coverslip.

## Organotypic culture and cell migration assay

After removing and washing the cerebellum, DNA constructs were injected between lobes and electroporated using the following parameters: 5 pulses, 30 V each, 50 ms ON-950 ms OFF (Harvard Apparatus ECM 830 BTX). After electroporation, the cerebellum was sliced into 280 µm sections using a tissue chopper machine (McIlwain Tissue Chopper 800 series Vibratome). The sliced cerebellum was placed in a 10 cm Petri dish with 6 ml of media and gently rocked to fully separate all the sections. Three to four slices were transferred to a filter insert (Millipore, Millicell Cat# PICM0RG50) with 1 ml SFM in the bottom well. Sections were incubated ON at 37°C and 5% $CO_2$. The next day the insert was transferred to a 35 mm glass-bottom dish (Mat Tek Cat# P35G-0–10C) for live image recording for 15 hr, using a Zeiss 510 META microscope.

## Transfection constructs

The RhoA constitutively active constructs were purchased from Addgene (pcDNA3-EGFP-RhoA-Q63L; Plasmid #12968). The ctrl-GFP construct was purchased from Lonza (pmaxGFP Vector). P75NTR-GFP construct was a kind gift from Dr Moses Chao.

## Transwell migration assay

Transwell assay was performed following the manufacturer's guidance (Costar Cat# 3422). Briefly, granule cells were plated in the top of the transwell insert 8 µm pore size (Costar Cat# 3422), at a final

density of $5\times10^5$ cells/insert. The total number of cells added in the top of the filter was the same for all conditions. The final volume for the top chamber was 100 and 600 µl for the bottom chamber. For control treatments, no ligand was added to the bottom chamber, for chemotactic analysis, BDNF was added to the bottom chamber to a final concentration of 50 ng/ml. The insert was incubated for 24 hr at 37°C and 5% $CO_2$. Cells were fixed using 4% ice-cold paraformaldehyde for 20 min and washed three times with PBS. Using a Q-tip, the cells on the top of the filter were removed by scraping the surface. Cells at the bottom of the filter were stained using 1 µg/ml of Dapi for 3 min at RT; 90% of the filter surface was imaged using a Nikon E1000 with a ×4 magnification. The total number of cells migrated was quantified using ImageJ version 1.51.

For the inhibition experiments, the specific inhibitor was added to the bottom and top compartments. All the inhibitors were added from the time of plating the cells and maintained for 24 hr. For RhoA inhibition, we use Rho kinase inhibitor Y27632 (Calbiochem Cat# 688000) to a final concentration of 1, 10, and 50 µM.

## In vivo migration assay

P7 mouse pups were injected with 50 mg/kg EdU intraperitoneally. After 24 or 48 hr, animals were deeply anesthetized with ketamine/xylazine and perfused with 4% PFA/PBS. Brains were removed and postfixed in 4% PFA/PBS overnight at 4°C, then cryopreserved with 30% sucrose. Sections (12 µm) were cut using a Leica cryostat and mounted onto charged slides. EdU stained was developed following the manufacturer's instructions (Thermo Fisher Scientific Cat# C10337). Stained for PAX6 was started immediately after finishing EdU staining, as described above. Sliced sections were imaged using a Nikon E1000 with a ×10 magnification. The total number of EdU+ and PAX6+ cells was quantified using ImageJ version 1.51.

## GTPase activation analysis

Cells were obtained from P7 WT and p75KO rat pups and cultured as described above for 48 hr. Cells were processed for G-Lisa analysis according to the manufacturer's instructions. G-Lisa for RhoA (Cytoskeleton Inc, #BK124).

## Western blot

Cultured cells were washed with ice-cold PBS and homogenized using 1% NP40, 1% Triton, and 10% glycerol in TBS buffer (50 mM Tris, pH 7.6, 150 mM NaCl) with a protease inhibitor cocktail (Roche Products, 11 836 153 001). Proteins were quantified using the Bradford assay (Bio-Rad 500-0006) and equal amounts of protein were run on SDS gels and transferred to a nitrocellulose membrane. To ensure equal protein levels, blots were stained with Ponceau before incubation with antibodies. The blots were then rinsed and blocked in 5% nonfat dried skim milk in TBS-T for 1 hr at RT. Blots were incubated with primary antibodies diluted 1:1000 in 1% BSA in TBS buffer overnight at 4°C. Primary antibodies used: anti-p75 (R&D AF367, RRID: AB_2152638, 1:500), PCNA (BD Bioscience 610664, RRID: AB_397991). The blots were washed with TBS-T three times 10 min each and incubated with Licor secondary antibody for 1 hr at RT. All secondary antibodies were diluted at 1:10,000. Membranes were washed three times 10 min each in TBS-T. The membranes were analyzed using Licor Odyssey infrared imaging system (LICOR Bioscience). To confirm equal protein levels, blots were reprobed for actin. All analyses were performed at least three times in independent experiments. Bands were quantified using ImageJ version 1.51s.

## Tissue clarification

The clarification of cerebellar slices was done following the iDISCO protocol (version 2016). Briefly, after the time-lapse experiment was finished, the membrane filter that contained the slice was cut using a scalpel blade. The section was fixed at RT for 1 hr and washed with PBS RT three times 20 min. The section was permeabilized with ×1 PBS/0.5% Triton X-100/20% DMSO at RT for 2 hr and blocked at RT using donkey serum 5% and BSA 1% in PBS for 2 hr. The primary antibody was incubated overnight at 4°C in a rocking station. The next day, the primary antibody was washed with PBS three times 30 min. The secondary antibody was incubated overnight at 4°C in a rocking station.

For the clearing procedure sample was dehydrated in ascending methanol series: 20%, 40%, 60%, 80%, 100%, 100%; for 1 hr each at RT. The sample was incubated in 66% dichloromethane (DCM,

Sigma 27099712 × 100 ml)/33% methanol at RT for 3 hr, with shaking, and then incubated in 100% DCM for two times 15 min to wash the MeOH, with shaking. Finally, the sample was incubated in dibenzyl ether (DBE, Sigma 108014-1KG). To acquire the images, the sample was placed in a glass-bottom 35 mm Petri dish covered with DBE.

## Statistical analysis

All statistical analyses were performed using GraphPad Prism 7.0. Experimental groups were compared using either Student's t-test, one- or two-way ANOVA followed by Tukey's post hoc analysis, as appropriate, $p < 0.05$ was considered significant. The specific statistical analysis is indicated in each figure legend. Error bars represent SEM.

## Acknowledgements

The p75-GFP construct was generously provided by Moses Chao (NYU Medical School). This work was funded by NIH/NINDS 1R56NS094589 to WJF, and Rutgers Busch Biomedical Grant AWD00009650 to JPZ.

# Additional information

## Funding

| Funder | Grant reference number | Author |
|---|---|---|
| Rutgers Busch Biomedical | AWD00009650 | Juan P Zanin |
| National Institute of Neurological Disorders and Stroke | 1R56NS094589 | Wilma J Friedman |

The funders had no role in study design, data collection and interpretation, or the decision to submit the work for publication.

## Author contributions

Juan P Zanin, Conceptualization, Data curation, Formal analysis, Funding acquisition, Methodology, Writing – original draft, Writing – review and editing; Wilma J Friedman, Conceptualization, Funding acquisition, Project administration, Writing – review and editing

## Author ORCIDs

Juan P Zanin ⬤ http://orcid.org/0000-0002-6216-0101
Wilma J Friedman ⬤ http://orcid.org/0000-0002-3638-3504

## Ethics

This study was performed in strict accordance with the recommendations in the Guide for the Care and Use of Laboratory Animals of the National Institutes of Health. All of the animals were handled according to approved institutional animal care and use committee (IACUC) protocols (201800118) of Rutgers University. Perfusion was performed under Ketamine/Xylazine anesthesia, and every effort was made to minimize suffering.

## Decision letter and Author response

Decision letter https://doi.org/10.7554/eLife.79934.sa1
Author response https://doi.org/10.7554/eLife.79934.sa2

# Additional files

## Supplementary files
• MDAR checklist

## Data availability

All data generated or analyzed during this study are included in the manuscript and supporting file; Source Data files have been provided for Figures 2 to 7. All RAW images for the western blot have also been provided as Source Data. The code for quantification of Figure 3 has been provided as Figure 3 - Source Code 1.

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
