## [Editor Report]

This study investigates a key molecular mechanism that drives neuronal migration. The results indicate that the p75 neurotrophin receptor provides an anti-migratory cue for granule cell precursors in developing cerebellum. This topic has been a topic of wide developmental interest that integrates previous and recent findings.

---

## [Decision Letter]

**Decision letter after peer review:**

Thank you for submitting your article "p75NTR prevents the onset of cerebellar granule cell migration via RhoA activation" for consideration by *eLife*. Your article has been reviewed by 3 peer reviewers, and the evaluation has been overseen by a Reviewing Editor and Jonathan Cooper as the Senior Editor. The following individual involved in the review of your submission has agreed to reveal their identity: Victor V Chizhikov (Reviewer #3).

Essential revisions:

There is general agreement among the reviewers that the mechanism of cerebellar granule neuron migration and its experimental support is considered to be rigorous and well done. The manuscript is of interest and the examination of the roles of p75 and RhoA in cell migration and cell cycle exit is appropriate.

However, several reservations were raised.

1) Given prior studies on this topic on neural progenitors such as granule cells, it is particularly important that the current results are discussed in the context of previously published roles of p75NTR to emphasize the significance and interpretation of the present findings. As an example, a discussion of these results with the role of p75NTR in programming and maintenance of a stem-like phenotype in mesenchymal stem cells and cancer cells may be informative (Vidal and Redmer, 2020).

2) In addition, there is a need to clearly address the reviewers' technical and conceptual concerns brought up in the reviews below.

*Reviewer #1 (Recommendations for the authors):*

I find the color choices in Figures 1, 2, and 3 problematic – in Figure 1A, the authors indicate that p75NTR staining is white, Ki67 is red, and Pax6 is green. Two issues – I find the white staining difficult to assess and the authors do not identify the source of the blue stain. Panel 1B, with p75NTR shown as red, provides considerably better contrast and resolution of the p75NTR expression pattern. Second, the blue nuclear staining is not identified in the panels or figure legends.

The fact that migratory rates differ between wild-type and p75NTR-/- CGPs in the absence of added ligand (Figure 5) and the earlier Yamashita results do not, on their own, indicate that p75NTR in this context is functioning in a ligand-independent manner. I agree that this should be put forward as a possibility but with the caveat that autocrine or paracrine ligands could have a role in either setting.

In the last paragraph – the authors state that 'A previous study proposed that proBDNF negatively regulates the migration of granule cell precursors and this effect is mediated by p75NTR. In that work, the author described that p75NTR-/- mice have reduced CGNs migration from the EGL'. A citation is required here.

*Reviewer #2 (Recommendations for the authors):*

1) This reviewer is having a hard time differentiating parts of the present manuscript from the group's previously published work where they already demonstrated the increased P75NTR expression in the proliferating granule cell progenitors. Furthermore, given the function of P75NTR (and RhoA) in the cell cycle exit, wouldn't some of the phenotypes observed here be secondary to that? The manuscript would greatly benefit from a thorough discussion on the context-dependent function of the P75NTR in the cell cycle exit and migration of the granule cells in the cerebellum.

2) As the authors mentioned in the parts of the manuscript, neurotrophin signalling and P75NTR have been implicated in neural migration as a pro-migratory que before. A detailed introduction and discussion of these findings and how they compare to the present study would be highly valuable. This brings me back to my first point; Could some of the present phenotypes be secondary to their effect on the cell cycle rather than directly on migration?

*Reviewer #3 (Recommendations for the authors):*

1. Lines 215-219 and Figure 5A, B. For clarity, I recommend explaining (in either the text or the figure legend) that the same number of granule cells was placed on the top of each insert. Thus, differences in cell density at the bottom filter reflect differences in migration.

2. The authors conclude that exogenous p75NTR prevents granule cell migration even in the presence of chemotactic signal BDNF. Shh stimulates the expression of p75NTR (Figure 2A, D). It would be interesting to test whether BDNF suppresses the expression of endogenous p75NTR. This experiment could provide an additional mechanism of how BDNF promotes the exit of GCs from the EGL.

3. Lines 289-290, 308. The rationale for conditionally deleting p75NTR in granule cell precursors is not clearly explained, likely because several lines were accidentally deleted in the manuscript. Can you please, check this paragraph? Contrary to what was stated, Figure 1 does not test for expression of p75NTR in Purkinje cells.

4. Figure 6E. The genotype of the mouse shown in this panel should be provided. Line 319. Should it be Figure 6E rather than Figure 6D? Contrary to what was stated, Figure 6E has no dotted line.

5. Lines 327-329. Based on the lack of EdU/Ki67+ cells in the IGL of Atoh1-Cre;p75f/f mice, the authors conclude: "indicating that no premature cell migration of proliferating GCPs was observed when we removed p75NTR, therefore cells still exited the cell cycle before migration." I found this conclusion somewhat misleading. If granule cells do not migrate prematurely, how can the increased number of labeled GCs in the IGL of Atoh1-Cre;p75f/f mice be explained? Do these cells migrate faster in the conditional mutant? I recommend modifying the original conclusion by simply stating that "loss of p75NTR does not prevent normal differentiation of GCs as they exit their proliferative niche in the EGL".

6. A control panel (a p75-GFP transfected slice without Y27632) is needed to appreciate the phenotype in panel 7F.

7. Lines 379-393. Cdc42 data are not well integrated into the main story of the paper, beyond showing that "After blocking Cdc42, the majority of GFP+ cells remained in the EGL and almost no radial migration towards the IGL was observed (Figure S2B, video 5), similar to the results observed after overexpressing p75NTR-GFP (see Figure 5E). I recommend that the authors either better link their Cdc42 data with the rest of their data (for example, by manipulating Cdc42 activity in p75NTR mutants and evaluating and discussing resulting migratory phenotypes) or delete the Cdc42 section.

---

## [Author Response]

Reviewer #1 (Recommendations for the authors):I find the color choices in Figures 1, 2, and 3 problematic – in Figure 1A, the authors indicate that p75NTR staining is white, Ki67 is red, and Pax6 is green. Two issues – I find the white staining difficult to assess and the authors do not identify the source of the blue stain. Panel 1B, with p75NTR shown as red, provides considerably better contrast and resolution of the p75NTR expression pattern. Second, the blue nuclear staining is not identified in the panels or figure legends.

We will modify the color code of the figures, and provide the individual channels in the supplementary figures.

The fact that migratory rates differ between wild-type and p75NTR-/- CGPs in the absence of added ligand (Figure 5) and the earlier Yamashita results do not, on their own, indicate that p75NTR in this context is functioning in a ligand-independent manner. I agree that this should be put forward as a possibility but with the caveat that autocrine or paracrine ligands could have a role in either setting.

We completely agree with the reviewer, we cannot discard autocrine or paracrine release of a migratory ligand, specially knowing that BDNF might be synthesized by the granule cells. We will emphasis this better in the discussion.

In the last paragraph – the authors state that 'A previous study proposed that proBDNF negatively regulates the migration of granule cell precursors and this effect is mediated by p75NTR. In that work, the author described that p75NTR-/- mice have reduced CGNs migration from the EGL'. A citation is required here.

We will correct the citation mistake. However, the mentioned paper has some contradictory results. In that paper the author suggests that proBDNF via p75NTR prevents the migration of the CGN. However, when they inhibited proBDNF they observed more migration. An increase in migration should then be observed if p75 is blocked, or in the absence of the receptor such as the p75NTR-/- mice, however the authors found fewer migrating cells in the p75NTR-/-.

Reviewer #2 (Recommendations for the authors):1) This reviewer is having a hard time differentiating parts of the present manuscript from the group's previously published work where they already demonstrated the increased P75NTR expression in the proliferating granule cell progenitors.

In the present work, we extend our previous findings regarding the expression of p75NTR in proliferating GCPs to support our main idea that p75NTR is expressed to prevent migration of the GCPs. We also expanded the relevance of p75NTR as a migratory inhibitor for CGN since overexpressing the receptor after the cells exit the cell cycle was not sufficient to make the cell re-enter the cell cycle.

Furthermore, given the function of P75NTR (and RhoA) in the cell cycle exit, wouldn't some of the phenotypes observed here be secondary to that? The manuscript would greatly benefit from a thorough discussion on the context-dependent function of the P75NTR in the cell cycle exit and migration of the granule cells in the cerebellum.

We agree, and we have added further discussion regarding this possibility. The reviewer raises a very interesting question, what is the connection between cell cycle (proliferation) and differentiation? We believe that the presence of p75NTR in the GCP contributes to maintaining the cells in the EGL, upon cell cycle exit, induced by different ligands, one of them being proNT-3, the differentiation process will start, reducing the expression of the receptor in this cell population. One of the open questions we are still persuing is how the exit of the cells from cell cycle will reduce the expression of p75NTR.

2) As the authors mentioned in the parts of the manuscript, neurotrophin signalling and P75NTR have been implicated in neural migration as a pro-migratory que before. A detailed introduction and discussion of these findings and how they compare to the present study would be highly valuable. This brings me back to my first point; Could some of the present phenotypes be secondary to their effect on the cell cycle rather than directly on migration?

We agree, and we have included further discussion about this possibility.

Reviewer #3 (Recommendations for the authors):1. Lines 215-219 and Figure 5A, B. For clarity, I recommend explaining (in either the text or the figure legend) that the same number of granule cells was placed on the top of each insert. Thus, differences in cell density at the bottom filter reflect differences in migration.

We will clarify this in the text.

2. The authors conclude that exogenous p75NTR prevents granule cell migration even in the presence of chemotactic signal BDNF. Shh stimulates the expression of p75NTR (Figure 2A, D). It would be interesting to test whether BDNF suppresses the expression of endogenous p75NTR. This experiment could provide an additional mechanism of how BDNF promotes the exit of GCs from the EGL.

In our previous work, we identified that only proNT-3 was capable of inducing cell cycle exit of the GCPs. Although we did not test whether BDNF suppresses the expression of p75NTR, BDNF was not able to induce cell cycle exit of the GCP in culture, for which we speculate that BDNF might not reduce p75NTR expression levels. The reviewer raises an interesting point about the dynamic cellular changes between these three cellular stages during development, proliferation, differentiation, and migration, although further studies are needed to dissect the specific changes the cell must undergo to transition between these stages.

3. Lines 289-290, 308. The rationale for conditionally deleting p75NTR in granule cell precursors is not clearly explained, likely because several lines were accidentally deleted in the manuscript. Can you please, check this paragraph? Contrary to what was stated, Figure 1 does not test for expression of p75NTR in Purkinje cells.

Thank you for pointing this out. 2 lines were deleted, we have incorporated them back into the text. Since Purkinje cells also express p75NTR, it was necessary to use the conditional knockout to investigate the specific role in granule cell progenitors.

4. Figure 6E. The genotype of the mouse shown in this panel should be provided. Line 319. Should it be Figure 6E rather than Figure 6D? Contrary to what was stated, Figure 6E has no dotted line.

Indeed, it should refer to Figure 6E. We will also clarify the genotype of the mice in the figures. Figure 6E has a dotted line although it is very thin, we will increase the line thickness.

5. Lines 327-329. Based on the lack of EdU/Ki67+ cells in the IGL of Atoh1-Cre;p75f/f mice, the authors conclude: "indicating that no premature cell migration of proliferating GCPs was observed when we removed p75NTR, therefore cells still exited the cell cycle before migration." I found this conclusion somewhat misleading. If granule cells do not migrate prematurely, how can the increased number of labeled GCs in the IGL of Atoh1-Cre;p75f/f mice be explained? Do these cells migrate faster in the conditional mutant? I recommend modifying the original conclusion by simply stating that "loss of p75NTR does not prevent normal differentiation of GCs as they exit their proliferative niche in the EGL".

We will clarify the text. By “premature” migration we were referring to cells that might be migrating even before they exit the cell cycle, which is not the case, in both genotypes (WT and p75FL, Atoh1Cre) the GCP exits the cell cycle before starting migrating.

6. A control panel (a p75-GFP transfected slice without Y27632) is needed to appreciate the phenotype in panel 7F.

A p75-GFP transfected slice without Y27632 is provided in Figure 5G, this is the first evidence we presented as p75NTR blocking the GCP migration. We consider it would be repetitive to add the same (or a similar) result in Figure 7, which is already an overcrowded figure.

7. Lines 379-393. Cdc42 data are not well integrated into the main story of the paper, beyond showing that "After blocking Cdc42, the majority of GFP+ cells remained in the EGL and almost no radial migration towards the IGL was observed (Figure S2B, video 5), similar to the results observed after overexpressing p75NTR-GFP (see Figure 5E). I recommend that the authors either better link their Cdc42 data with the rest of their data (for example, by manipulating Cdc42 activity in p75NTR mutants and evaluating and discussing resulting migratory phenotypes) or delete the Cdc42 section.

This has been a concern for two reviewers therefore we will delete this data. The importance of Cdc42 in GCP migration has already been published (Govek 2019), we were using this data to validate our organotypic migratory model.